# An Efficient Routing Scheme Based on Node Attributes for Opportunistic Networks in Oceans

**DOI:** 10.3390/e24050607

**Published:** 2022-04-27

**Authors:** Lige Ge, Shengming Jiang

**Affiliations:** College of Information Engineering, Shanghai Maritime University, Shanghai 201306, China; smjiang@shmtu.edu.cn

**Keywords:** opportunistic networks, node attributes, delivery competency, forwarding willingness

## Abstract

Along with the fast development of the marine economy and ever-increasing human activities, handy and reliable marine networking services are increasingly required in recent years. The ocean faces challenges to support cost-effective communication due to its special environments. Opportunistic networks with easy deployment and self-curing capability are expected to play an important role to adapt to such dynamic networking environments. In the literature, routing schemes for opportunistic networks mainly exploit node mobility and local relaying technologies. They did not take into account the impact of node behaviors on encountering opportunities and in case of no further relaying, network performance would be greatly degraded. To solve the problem, we propose an efficient routing scheme based on node attributes for opportunistic networks. We first construct delivery competency to predict the further relay nodes. Then a forwarding willingness mechanism is introduced to evaluate the relaying probability combining device capacity and movement behaviors of nodes. Finally, the utility metric is used to make decisions on message forwarding. The results show that the proposed scheme improves network performance in terms of delivery ratio, average latency, and overhead ratio as compared to other schemes.

## 1. Introduction

The rapid development of the marine economy and ever-increasing human activities in oceans require reliable and cost-effectiveness networking services. Although terrestrial Internet can be easily accessed almost anytime and anywhere, the ocean faces challenges to support satisfied services due to huge differences with terrestrial environments in terms of geographical location, climatic conditions, and user distributions. The satellite can provide reliable Internet services in oceans; however, its expensive services make it difficult to become popular especially among ordinary users [1]. In such challenging environments, opportunistic networks (OppNets) are expected to play an important role, due to its easy and quick deployment without fixed infrastructure, forming dynamic networks by constant mobility of nodes.

In OppNets, no prior knowledge is made with regard to the existence of a complete route between two nodes wishing to communicate. The communications of source and destination mainly rely on frequent mobility of nodes to create opportunistic contacts. Figure 1 is an illustration of routing in OppNets. In Figure 1, there is no direct route from source *S* to destination *D*. At 14:00 p.m., node *S* forwards the packets to node 1; at 15:10 p.m., the carrier node 1 forwards the packets to node 4; the destination *D* receives the packets from node 4 at 16:00 p.m. Data packets from node *S* are eventually delivered to node *D* even if they might never be connected to the same network, at the same time [2]. This is due to the fact that the movement of intermediate nodes helps to forward the packets during delivery. However, the questions of how data forwarding of nodes is routed to the destination at low cost, and how to guarantee robust communication in the face of frequent changing networks still present major challenges in dynamic marine environments.

Many studies have been carried out on routing schemes [3,4,5]. These routing schemes, based on dissemination and context technologies, can fulfill packet transmissions. However, most of them are based on an end-to-end connection and the fixed paths before packet transmissions, so they are not suitable for OppNets. Furthermore, the current node makes decisions on message delivery only dependent on local forwarders without consideration of further intermediate nodes. The question of whether the relay nodes with communication function have enough capability to participate in the message relay is ignored. In this case, the successful delivery ratio decreases and unnecessary network resources waste, especially in dynamic marine networking environments.

In this paper, we propose an efficient routing scheme based on node attributes for opportunistic networks (RSNA) in oceans. The major contributions of the paper are as follows.

(1) The delivery competency is constructed to predict the existence of the further intermediate nodes.

(2) We design a forwarding willingness mechanism based on device capacity and movement behaviors to qualify whether the nodes can participate in the delivery process.

(3) By using a utility metric combining delivery competency and forwarding willingness, we developed the RSNA, an efficient routing scheme based on node attributes in opportunistic networks to improve network performance.

The rest of the paper is organized as follows. Section 2 introduces the related works. Section 3 gives a detailed description of the proposed scheme. The performance evaluation is given in Section 4. Finally, we conclude briefly the paper in Section 5.

## 2. Related Works

In OppNets, the network scale and topology information cannot be known in advance. Packets are transmitted determined by the opportunity of encounter between nodes in OppNets. Currently, opportunistic networks are mainly applied to specific fields such as wildlife tracking [6], disaster rescue [7], and vehicle-mounted monitoring networks [8]. The maritime opportunistic network introduces the opportunistic network into the marine environment, and it has both the characteristics of the opportunistic network and the marine environment [9]. In the maritime opportunistic network, vessels are regarded as mobile networking nodes, and packet transmission depends on the opportunistic contacts between vessels.

An appropriate routing scheme for opportunistic networks can provide reliable message transmissions. In recent years, there are many routing schemes for OppNets in the literature, which are summarized as follows.

### 2.1. Copy-Based Routing

The copy-based routing strategy generates a certain number of message copies in opportunistic networks to improve the success delivery ratio of the destination. Direct transmission [10] utilizes the single-copy strategy to deliver directly the packets from source to its destination, not relying on any additional intermediate nodes. In this algorithm, all packets are delivered only once, and thus have minimal network overhead. However, it has a higher delay and great risks of packet loss. Epidemic routing [11] produces several copies of the information among all the neighbors in each hop, and copies action continues until the information delivers to the destination, and the process is successfully delivered with lower latency. However, too many copies can cause broadcast traffic, which in turn significantly degrades network performance. Compared to Epidemic routing, SprayandWait routing [12] distributes a few message copies into the network, and then routes each copy independently toward the destination, which performs significantly fewer transmissions and lower average delivery delays.

### 2.2. Active Movement-Based Routing

In active movement-based routing mechanisms, some ferrying nodes provide communication services through active movement. In [13], a mobility-assisted approach is proposed, which utilizes a set of special mobile nodes called message ferries to communicate for nodes in a sparse network. Reference [14] presents a message ferrying (MF) scheme to exploit controlled mobility for data transmission in delay-tolerant networks. However, some prior knowledge is assumed of networks of stationary nodes and known traffic demands.

### 2.3. Utility-Based Routing

The utility-based routing strategy selects the appropriate next hop node to avoid blind forwarding of messages. An energy-aware social-based routing scheme in opportunistic networks is proposed [15]. In this scheme, energy awareness as an important criterion in the routing decision is introduced. The result shows that the total life span of the OppNets is increased. However, delivery costs have been significantly improved. Ref. [16] designs an energy-efficient routing protocol named EHBPR for infrastructure-less OppNets, which utilizes multi-factor constraints to reduce the number of packets transferred in the network and energy consumption of nodes. In [17], an energy-based routing protocol for OppNets is proposed, which performs a genetic algorithm on the personal information about a node to select an appropriate node as the next hop. However, the limitation is that the movement model used in this work is highly suited to human mobility scenarios. Prophet [18] introduces a probabilistic metric called delivery predictability, a decision is made on whether or not to forward the message to this node. Reference [19] proposes a probabilistic routing scheme based on game theory (PRGT) to stimulate cooperation among selfish nodes. However, one major problem with OppNets is how to find and solve the problem of gangs defrauding the bounty.

As mentioned earlier, the copy-based routing scheme improves the delivery ratio; however it consumes a lot of resources and degrades network performance due to multiple message copies. In active movement-based routing, special nodes with strong mobility and capability to store messages are introduced to assist the data transmission. However, the assumption is made with fixed destination node location and the special nodes significantly differ from ordinary nodes. The utility-based routing strategy mainly relies on some metrics, such as the energy, delivery probability, etc., to perform the routing strategies. However, the problems are not considered that the case of no further forwarding nodes may occur and how node attributes affect forwarding opportunities in oceans.

## 3. The Proposed Scheme: RSNA

### 3.1. Motivation

In maritime opportunistic networks, vessels acting as networking nodes are randomly deployed over the sea. The communication links are significantly affected by sparse scattering, sea wave movement, and the ducting effect over the sea surface [20], due to the extreme complexity of the marine environment. Therefore, the impact of marine environmental factors on networking communication cannot be ignored. In this paper, we introduce the marine channel propagation model mentioned in our previous work [21] into opportunistic networks for data transmission.

Routing and forwarding issues are the main focus for communication in OppNets; that is, it is necessary to find the desirable route to the destination. As shown in Figure 2, assuming that source node is S, the destination node is D. Nodes N1, N3 are the closet nodes of S toward the destination, they will relay the packets. After this hop, N1 and N3 get higher priority to be the relays. However, some seemingly qualified nodes of N1 and N5 of N3’ next hop have no further available candidates to relay the packets, and only end up dropping the packets in a time period. The node N8 is similar to this. Actually, if node N2 gets the chance to deliver the packets, node D can eventually receive the packets through the route N2-N4-N7. The route requests of some seemingly qualified nodes may cause the packets to be delayed indefinitely and greatly degrade network performance. Therefore, solving the above problems is our target in this paper.

### 3.2. The Description of RSNA

The basic idea of how RSNA efficiently forwards messages is divided into three parts: delivery competency, forwarding willingness mechanism, and utility-based forwarding.

#### 3.2.1. Delivery Competency

An appropriate wireless channel is fundamental to construct efficient communication, which is one of the strongest guarantees for data transmission. In our previous work [21], we have been proposed link availability prediction based on machine learning for opportunistic networks in oceans. Therefore, probabilistic link availability is directly used in this paper to qualify the probability for packet transmissions. If the node *r* can receive a message from node *l*, the link availability ($p_{la}$) can be defined by
(1)$$p_{la}=\left\{\begin{array}{cc} p_{lr}\;\mathrm{node}\;r\;\mathrm{has}\;\mathrm{received}\;\mathrm{the}\;\mathrm{message}\;m & \\ 0\;\mathrm{otherwise} \end{array}.\right.$$

Moreover, the encountering probability is used to describe the delivery relationship between node pairs, which can be given by    
(2)$$p_{l,r}=\frac{e_{l,r}}{\sum_{c=1,c\neq l}^{n}e_{l,c}}$$
where $e_{l,r}$ is the number of encounters between nodes *l* and *r*, and $e_{l,c}$ is the total number of encounters between nodes in a time period *T*.

The encountering process can be viewed as a binary set including discrete node and time period, whose stochastic process {x(t),tϵT} can be defined by the Markov chain, the state transfer for any state node {1,2,…,l,r,…,n} at any time {1,2,…,T}, there is
(3)$$\begin{array}{cc} p_{l,r} & =\{x(t+1)=r|x(t)=l,x(t-1)=l-1,\ldots ,x(1)=1\}\\ \; & =\{x(t+1)=r|x(t)=l\} \end{array}.$$

Delivery competency of nodes can be described by
(4)$$P_{FC}=\alpha p_{la}+\beta p,$$
where $p_{la}$ is the link availability, *p* is the encountering probability, and α and β are weights on delivery competency, respectively. In addition, α + β =1.

#### 3.2.2. Forwarding Willingness Mechanism

Forwarding willingness consists of the device capacity and movement behaviors of nodes. We define it into two categories as follows.

One category is related to the energy and buffer of node called device capacity (DC), which can be given by
(5)$$DC={\left(\lambda \cdot \frac{B_{r}\left(t\right)}{B_{init}}+\mu \cdot \frac{E_{r}\left(t\right)}{E_{init}}+1\right)}^{-1},$$
where $B_{r}$ and $B_{init}$ are residual and initial buffer size of node. Similarly, $E_{r}$ and $E_{init}$ are residual and initial energy of node at time *t*. λ and μ are influence factors of buffer and energy on device capacity, respectively.

The second category involves the movement behaviors of nodes (BN), which means the more movement relations of nodes, the higher forwarding probability. BN is defined as follows:(6)$$\begin{array}{c} BN=\frac{O_{l}}{I_{l}+O_{l}}\;\\ \left\{\begin{array}{c} I_{l}=\frac{\sum N_{l-1}}{\sum_{t\epsilon T}N_{t}}\;\\ O_{l}=\frac{\sum N_{l+1}}{\sum_{t\epsilon T}N_{t}} \end{array}\right. \end{array},$$
where incoming degree ($I_{l}$) is defined as the ratio of the number of upstream nodes (l−1) that forward message directly to itself to the total number of nodes. Similarly, the outgoing degree ($O_{l}$) represents the ratio of the number of direct downstream nodes (l+1) that receive forwarded message to the total number of nodes. Here, a forwarding willingness metric FW is given by
(7)$$FW=\zeta \log_{2}\left(DC+1\right)+\eta \log_{2}\left(BN+1\right),$$
where ζ and η are the influence factors, respectively.

#### 3.2.3. Utility-Based Forwarding

Combining node attributes, the utility-based forwarding is described by
(8)$$\begin{array}{cc} \; & U\left(l,m\right)=P_{FC}\left(l\right)\cdot {\left(FW_{l}\cdot m\cdot R_{l}-Tr_{l}^{m}\right)}^{-1}\\ \; & U\left(r,m\right)=P_{FC}\left(r\right)\cdot {\left(FW_{r}\cdot m\cdot R_{r}-Tr_{r}^{m}\right)}^{-1} \end{array},$$
where $R_{l}$ and $R_{r}$ are the resource values of forwarding messages for source/carrier node *l* and relay node *r*. $Tr_{l}^{m}$ and $Tr_{r}^{m}$ denote the costs of transmitting message *m* of node *l* and node *r*, respectively. Simultaneously, the utility metric needs to satisfy
(9)U(r,m)⩾U(l,m)+δ,
where δ is threshold, the value of which can differ depending on network requirement and design goal. The utility metric is made by joining the delivery competency and forwarding willingness evaluation together to make decisions on the relay node selection. Here, we not only predict the intermediate nodes of the current node *l*, but also whether the relay node *r* has the next hop. If link availability $p_{la}\left(l,r\right)$ exists, it implies that the node *l* has intermediate nodes. Furthermore, the prediction of FW(r) is used to evaluate the device capacity of the relay node *r* and the existence of its further relay nodes. The predictions of further relay nodes effectively diagnose the unsuitability of the seemingly qualified nodes, thus avoiding the unnecessary route requests and waste of network resources.

### 3.3. Routing Decision Scheme

In OppNets, an efficient routing scheme not only improves the success delivery ratio of message forwarding, but also degrades the overhead of the network. To promote our expression below, we utilize $F_{N_{n}}$ to denote the neighbor sets of node $N_{n}$, and  $F_{N_{n}}^{m}$ indicates the message sets carried by node $N_{n}$. In Algorithm 1, if the link between sender node $N_{l}$ and its neighbor node $N_{r}$ is available and node $N_{r}$ ($N_{r}$ϵ$N_{n}$) is destination node $N_{d}$, the messages are directly transferred from node $N_{r}$ to node $N_{d}$ (Algorithm 1: line 4). Otherwise, if the link availability of relay node $N_{r}$ and its neighbor $N_{rf}$ is greater than 0 (Algorithm 1: line 6), it is implied that node $N_{r}$ has a next hop, which identifies the unavailability of node $N_{1}$ without further relay nodes to the destination node in the example of Figure 2. Moreover, the forwarding willingness mechanism is further introduced to qualify whether the nodes can participate in the packet delivery process. When the forwarding willingness between node $N_{l}$ and node $N_{r}$ meets the communication demands, the value of $FW(N_{r},N_{rf})$ is further predicted (Algorithm 1: line 8). In $FW(N_{r},N_{rf})$, the prediction of $DC(N_{r},N_{rf})$ can abort some seemingly qualified nodes due to the insufficient of device capacity, and combining the movement behavior of $BN(N_{r},N_{rf})$ can jointly detect the unreachability of nodes $N_{5}$ and $N_{8}$ to the destination node in the example of Figure 2. This is because the nodes periodically exchange the metrics with each other during message routing so that nodes are available when they have an opportunistic contact. Then the Equation (Equation 9) is used to make decisions on whether the messages forward to the relay node $N_{r}$ (Algorithm 1: line 9). Finally, the messages are delivered hop-by-hop until the destination node is in proximity, and eventually to the destination node itself.
**Algorithm 1** RSNA routing algorithm
Begin 1:**for** each message m ϵ$F_{N_{n}}^{m}$ **do**2:   **for** each node $N_{l}$ϵ$F_{N_{n}}$ **do**3:     **if** the link between sender $N_{l}$ and neighbor node $N_{r}$ is available and $N_{r}$ = $N_{d}$ **then**4:        $N_{r}$ transmits message *m* to $N_{d}$5:     **else**6:        **while** $p_{la}\left(N_{r},N_{rf}\right)$> 0 **do**7:          **if** the value of $FW(N_{l},N_{r})$ meets the communication demands **then**8:             predict the value of $FW(N_{r},N_{rf})$9:             **if** $U(N_{r},m)$≥$U(N_{l},m)$ + δ **then**10:               node $N_{l}$ forwards message *m* to node $N_{r}$11:             **end if**12:          **end if**13:        **end while**14:     **end if**15:   **end for**16:**end for**
End


## 4. Performance Evaluation

We run the simulation with the opportunistic network environment (ONE) simulator [22] to evaluate the performance of proposed routing scheme (RSNA). In simulation scenarios, we assume that all nodes are not selfish or malicious to exchange messages with each other. The random movement model of nodes and intelligent link prediction are based on reference [21]. The simulation parameters are shown in Table 1.

We compare the performance of the RSNA with some classic routing schemes, including Prophet, Epidemic and SprayAndWait. The following metrics are used for performance analysis:

(1) Delivery ratio: the ratio of the number of packets delivered to the destination to the number of packets sent by the source.

(2) Average latency: the average time taken to transmit the number of packets from source to destination node.

(3) Overhead ratio: the ratio of the total number packets generated by source nodes to the total number packets forwarded by all nodes.

### 4.1. RSNA Parameters

As shown in Equations (Equation 4) and (Equation 5), the parameters of α, β, and μ and λ have a direct influence on routing decisions. We define 10 combinations with different values, as shown in Table 2.

From Figure 3, we can see that performance of packet delivery ratio and overhead ratio of combination ♯10 is the worst. This is because it gives all weights to a metric without considering any other information, which is not enough to make decisions on relay node selection. We can observe that the performance improves significantly when the multi-metrics are considered (β≠0 and λ≠0). Furthermore, the best performance of delivery ratio and overhead ratio is reached when the value of α (μ) and β (λ) is combination ♯6. In addition, the values of ζ and η have no significant impact on performance, so we set them to 0.5 and 0.5, respectively. They are also used in the experiment by default unless specifically mentioned.

### 4.2. Impact of Different Node Density on Routing Performance

In this section, we compare the RSNA routing scheme with other routing schemes under different node density. The results and analysis are shown as follows.

Figure 4a shows that the delivery ratio of all schemes improves as the node density increases accordingly, because more nodes implies more chances of forwarding, and thus higher delivery ratio. Due to the whole network flooding strategy with redundant replications, Epidemic has the lowest packet delivery ratio. The partial flooding of SprayAndWait and the relay selection mechanism of Prophet can effectively reduce the number of generated packets and achieve a higher packet delivery ratio than Epidemic’s blind forwarding. However, they are not taken into consideration the delivery competency of further intermediate nodes and forwarding willingness based on device capacity and node behaviors, so the delivery ratio improves more slowly than RSNA.

From Figure 4b, we can see that Prophet and SprayAndWait have a long average latency, because message carriers have to wait for an appropriate intermediate node for forwarding messages. Because the Epidemic scheme is based on flooding forwarding strategy without the relay node selection, it has a slightly higher latency than RSNA. The proposed RSNA scheme reduces the number of aborted messages due to the prediction of insufficient relay nodes and evaluation of relay possibility. Therefore, it has the lowest average latency.

Figure 4c shows the comparison of overhead ratio. As expected, Epidemic has the highest overhead, because the carrier of the same message will increase as the node density improves. In Prophet, it has more opportunities to choose an appropriate relay of higher delivery probability when the number of nodes increases, and thus has a large overhead ratio. Each message has a fixed cope in SprayAndWait, so the overhead is stable when the node density increases. However, the carrier of RSNA can select an efficient forwarder via the utility metric based on node attributes during message relaying, so RSNA has a lower overhead ratio than the other three algorithms.

### 4.3. Impact of Different Buffer of Nodes

In this section, we consider the impact of buffer size on the delivery ratio, average latency and overhead ratio, respectively. We set the buffer size of the nodes from 10 MB to 100 MB, the results are shown in Figure 5.

From Figure 5a, we can see that when the buffer size increases, the delivery ratio of all four schemes increases accordingly. For RSNA, with the energy and buffer limited, we construct the delivery competency to predict the further relaying, and employ the forwarding willingness to jointly implement the message delivery decisions. Therefore, the delivery ratio of RSNA is highest. Compared with RSNA, the other three algorithms do not consider node attributes, so the delivery ratio is relatively lower.

Figure 5b shows the comparison of average latency. Although the average latency of the four routing schemes decreases as the buffer size increases, the RSNA has the least average latency. For nodes in networks, the larger the buffer size, the more message copies can be retained, and the lower average latency. In Prophet, the nodes have to wait for an opportunity of higher delivery probability for message forwarding, thus the latency is the highest. Similarly, in Epidemic and SprayAndWait, the carrier nodes also need to wait for helper nodes, which increases the latency of message forwarding.

As shown in Figure 5c, the overhead of RSNA is similar to SprayAndWait, much lower than that of Epidemic and Prophet. In SprayAndWait, the replications of any message are predefined, so the overhead ratio is also very low. Because the forwarding mechanism is based on flooding in Epidemic, it has a higher overhead than the other three routing algorithms.

## 5. Conclusions

The paper designs a novel routing scheme based on node attributes for OppNets in oceans. The scheme first constructs the delivery competency to predict the existence of further relaying. Then a forwarding willingness mechanism of device capacity and node behaviors is introduced to qualify the probability that the nodes can participate in the delivery process. Finally, the utility metric is proposed to make decisions on message delivery. The results show that the proposed scheme improves network performance in terms of delivery ratio, average latency, and overhead ratio as compared to other schemes. In future work, the routing scheme will further consider the impact of packets dropped from the cache on system performance and develop the exploration of further relay nodes toward the destination as soon as possible to reduce waste of resources and network overhead.

## Figures and Tables

**Figure 1 entropy-24-00607-f001:**
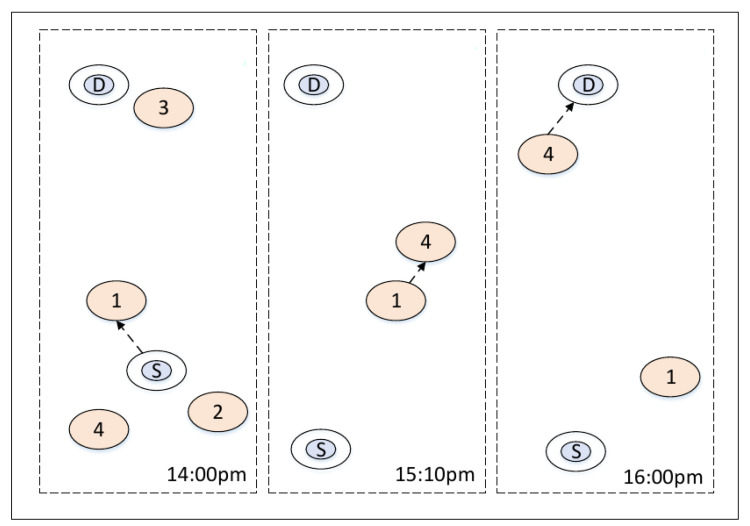
Illustration of routing in opportunistic networks.

**Figure 2 entropy-24-00607-f002:**
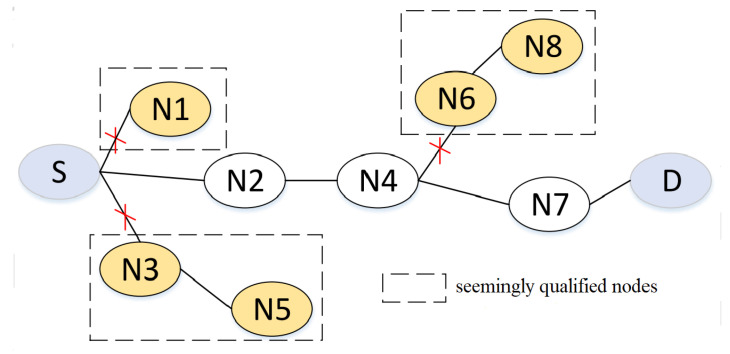
Relay node selection example.

**Figure 3 entropy-24-00607-f003:**
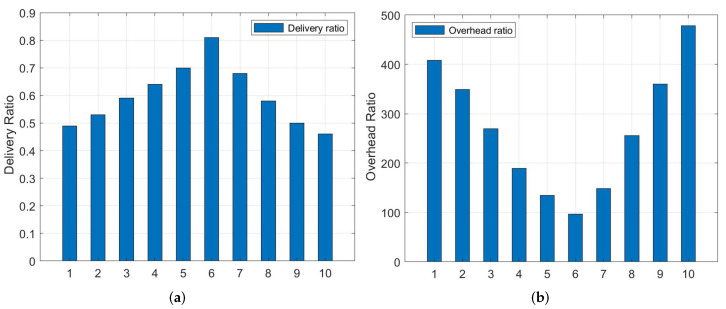
(**a**) Delivery ratio vs. 10 coefficient combinations. (**b**) Overhead ratio vs. 10 coefficient combinations.

**Figure 4 entropy-24-00607-f004:**
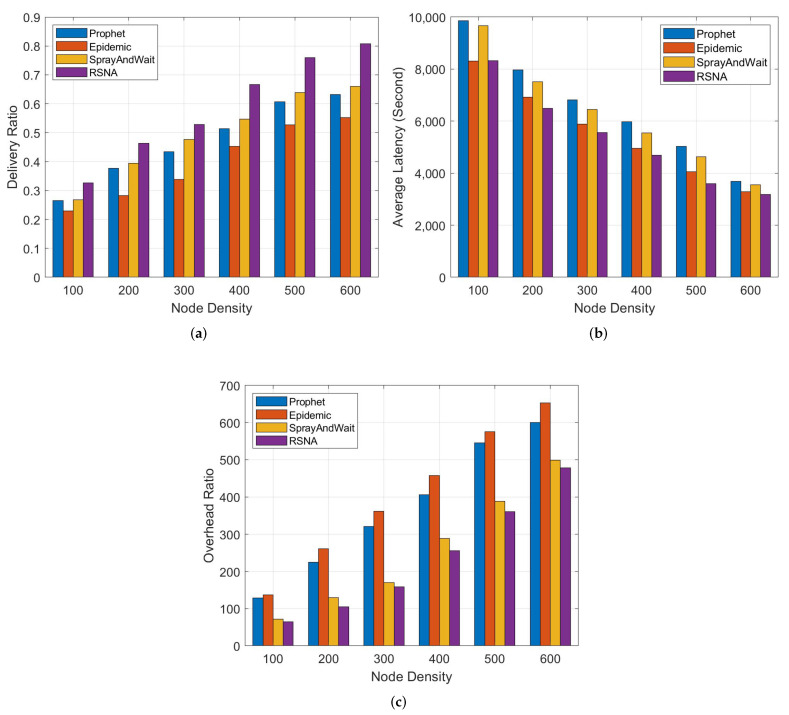
Comparison of different schemes. (**a**) Delivery ratio vs. node density. (**b**) Average latency vs. node density. (**c**) Overhead ratio vs. node density.

**Figure 5 entropy-24-00607-f005:**
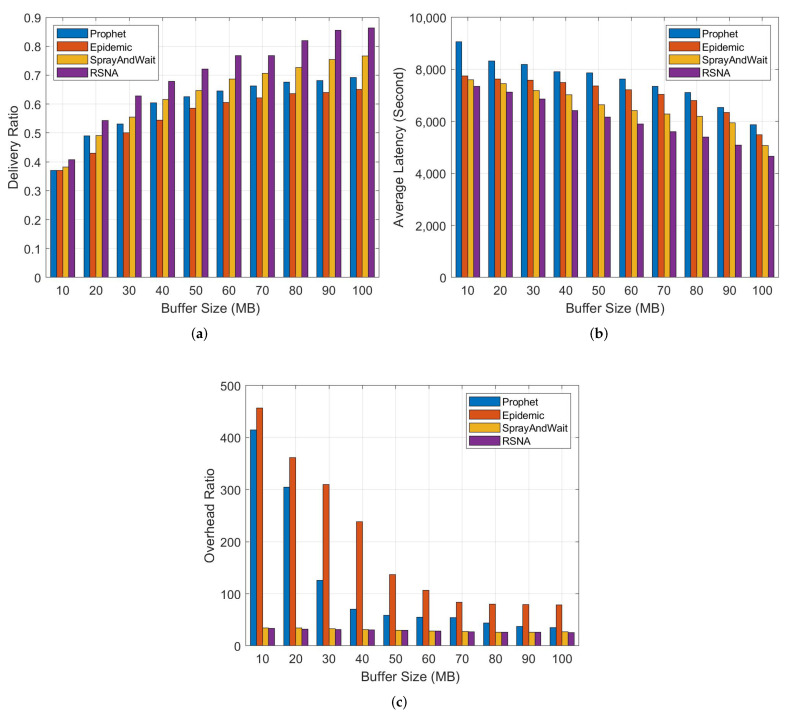
Comparison of different schemes. (**a**) Delivery ratio vs. buffer size. (**b**) Average latency vs. buffer size. (**c**) Overhead ratio vs. buffer size.

**Table 1 entropy-24-00607-t001:** Simulation parameters.

Parameter	Value
Transmission speed	250 kbps
Message size	500 KB–1 MB
Message interval	25 s–35 s
TTL	300 min
Node speed	0–30 Km/h
Number of groups	6
Initial energy	5000 J
Simulation time	43,200 s

**Table 2 entropy-24-00607-t002:** RSNA parameters.

Combination ♯	α (μ)	β (λ)
1	0.1	0.9
2	0.2	0.8
3	0.3	0.7
4	0.4	0.6
5	0.5	0.5
6	0.6	0.4
7	0.7	0.3
8	0.8	0.2
9	0.9	0.1
10	1.0	0

## Data Availability

The program code used in the research can be obtained from the corresponding author upon request.

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
