# Peer review of "An Efficient Routing Scheme Based on Node Attributes for Opportunistic Networks in Oceans"

_entropy, 2022, doi:10.3390/e24050607_

Round 1

Reviewer 1 Report

The paper proposes a routing scheme for OppNets which by enhancing the utility metrics used for forwarding the messages, improves the performance of the network. The routing scheme uses a forwardness willingness mechanism and delivery competency measure together as the utility metric. The results show improved performance on all metrics of the OppNet network. 

The paper is written well and clear to understand. The scenarios are explained clearly and the simulations parameters are explained well. 

  1. Do the nodes transfer willingness metric as a part of meta information exchanged when nodes meet? For example, nodes in Prophet exchange delivery predictability vectors or neighbor tables when they meet. Is RSNA doing the same as well?   If yes, please mention it in the paper in the algorithm or in the text.
  2. In the literature, please mention related works related to enery bassed routing schemes in OppNets. There are papers that consider energy based utility factors in OppNets. 
  3. As the title of the paper mentions OppNets in oceans, related works on this section is essential in my opinion. A subsection in the related literature regarding this topic needs to be included.
  4. The abstract doesn't mention anything about Oceans as well. This undermines the use case for the paper.
  5. Also, underwater networks have different transmission capabilities and restrictions compared to overground networks. Though OppNets can be a general theme for both overground and underwater networks, the architecture is quite different because of the communication challenges. How the authors visualize the routing scheme for OppNets in Oceans need to be explained in the paper.
  6. Figure 3(b) shows that RSNA has lower or similar latency like Epidemic. Is this because the routing path is known apriori? I am just not very sure on how RSNA achieves this improvement. Could you please clarify it as a reply to the comment?
  7. What is the traffic generation rate for the experimental setup? Do you also packet drops from the cache happening in your experiment?

Author Response

Dear reviewer,

We thank the reviewers for the time and effort that they have put into reviewing the previous version of the manuscript. The suggestions have enabled us to improve our work. We have considered these comments carefully and tried our best to address every one of them. Based on the instructions provided in your letter, we uploaded the file of the point-by-point response in Word. And the responses are marked in red color. Please see the attachment.

We appreciate for Reviewer’ warm work earnestly, and hope that the correction will meet with approval.

Once again, Thank you very much for the excellent and professional revision of our manuscript.

Reviewer 2 Report

After read the paper for one week, I have some suggestions:

1.Abstract: The abbreviation (OppNets) should not appear in abstract.

2.The main contribution of the paper is use prediction method to predict the further relay nodes, However, the main prediction mode is not appear in the paper.

3.The paper use simulation method to prove the result is better than traditional method, But in Table 1, if change the value of parameters (such as alpha, beta and lambda), is the results will be the same? Also, if the author could, please polt the flowchart of the contribution in software.

4.The conclusion seems missing the limitation. Please recheck.

5.Please use third person to replace the first person: for example

   In this paper, we design a….(page 9, line 216)…The paper designs…

   We first construct the…. (page 9, line 217) the paper first construct the….

   we will develop the….(page 9, line 222), the paper will develop the…

Author Response

(The authors gave the same response as above.)
